# Analysis of Predicted Amino Acid Sequences of Diatom Microtubule Center Components

**DOI:** 10.3390/ijms241612781

**Published:** 2023-08-14

**Authors:** Darya P. Petrova, Alexey A. Morozov, Nadezhda A. Potapova, Yekaterina D. Bedoshvili

**Affiliations:** 1Limnological Institute, Siberian Branch, Russian Academy of Sciences, Irkutsk 664033, Russia; 2Institute for Information Transmission Problems (Kharkevich Institute) of the Russian Academy of Sciences, Moscow 127051, Russia

**Keywords:** diatoms, microtubule organizing center, gamma-complex protein, Aurora A, centrins, posttranslational modification

## Abstract

Diatoms synthesize species-specific exoskeletons inside cells under the control of the cytoskeleton and microtubule center. Previous studies have been conducted with the visualization of the microtubule center; however, its composition has not been studied and reliably established. In the present study, several components of MTOC in diatoms, GCP (gamma complex proteins), Aurora A, and centrins have been identified. Analysis of the predicted amino acid sequences of these proteins revealed structural features typical for diatoms. We analyzed the conserved amino acids and the motives necessary for the functioning of proteins. Phylogenetic analysis of GCP showed that all major groups of diatoms are distributed over phylogenetic trees according to their systematic position. This work is a theoretical study; however, it allows drawing some conclusions about the functioning of the studied components and possible ways to regulate them.

## 1. Introduction

Diatoms are a large group of the unicellular autotrophic organisms that belong to the kingdom Chromista [1]. These microorganisms are capable of synthesizing species-specific silica exoskeletons (frustules). Modern diatom systematics is based on phylogeny of 18S rRNA and rbcL marker genes [2,3,4] and the morphology of silica frustules [5,6]. Phylum Bacillariophyta consists of classes Coscinodiscophyceae, Mediophyceae, and Bacillariophyceae (with subclasses Bacillariophycidae, Fragilariophycidae, and Urneidophycidae) and includes more than 18,000 species with variable morphology [6].

The main direction of diatom study is currently focused on a search of cell and genetic mechanisms providing various symmetry and structure of diatom silica frustules. It was shown that frustule morphogenesis control is carried out by the cytoskeleton; to date the microtubule significance is the most studied [7,8,9]. Back in the 70s of the twenty century, the formation of frustules with structure aberrations was shown in experiments with microtubule inhibitors [10,11,12,13,14,15,16]. The tubulin was localized as an underlying layer during the formation of the frustule [7] and some specialized structures [13,14]. It is assumed that diatom frustule symmetry is determined by the microtubule center. This structure is a homolog of the centrosome and involved in diatom cell division. The location and presence of a raphe on the frustule, a distinctive feature of pennate diatoms, depends on the location of the microtubule center [17,18,19]. For some species, it was shown that the development of specific structure rimoportula is under the influence of the labiate process apparatus that has a tightly ontogenetic relationship with the postmitotic microtubule system [17]. 

Microtubule functions and tubulin polymerization/depolimerization are regulated by a large number of proteins [20,21]; their nucleation occurs in the specific center of the microtubule organization [22]. The term MicroTubule Organizing Center (MTOC) was suggested by J.D. Pickett-Heaps [23] and includes morphologically distinct microtubule nucleation machinery at different species, both centrosomes and spindle pole bodies. The MTOC of a large group of organisms has various morphology and composition [20,21]. Diatoms, as plants and fungi, have the acentrosomal microtubule nucleation center, that is, their MTOC is not organized into triplets but rather has an amorphous morphology [24,25,26]. 

The eukaryotic MTOC is based on the γ-tubulin with the Gamma-tubulin Complex Proteins (GCP) [27,28]. Currently, these proteins are the most studied in various organisms (Table 1). It is supposed that MTOC morphology depends on the GCP composition. The most important components are GCP2 and GCP3, whereas GCP4, GCP5, and GCP6 are not found in all organisms and are not able to form a nucleation center. 

Earlier, the presence of α-, β- and γ-tubulin in the diatom genomes was shown [26,29,30]. The phylogenetic analysis of the known diatom tubulin sequences that had been identified using whole genome sequencing and MMETSP transcriptome database revealed that diatom tubulins divided into clades according to the main taxa of phylum Bacillariophyta [31]. Earlier, the analysis of *Thalassiosira pseudonana* and *Phaeodactylum tricornutum* genomic data revealed genes encoding the GCP2 and GCP3, as well as the centrins and protein kinase Aurora A [26].

In this study, we conducted a search for the nucleotide sequences of several MTOC components in well-known databases organized and curated by genomic and transcriptomic projects on the identification of these components in diatoms (according to the scheme shown in Figure 1).

The analysis of the predicted amino acid sequences made it possible to reveal the features of their structure and putative pathways of its regulation.

## 2. Results and Discussion

The search in the well-known databases revealed the next nucleotide sequences of diatom: GCP2-6, protein kinase Aurora A, and centrins. Moreover, nucleolin homolog sequences were identified; however, these data were included in another study. GCP8, GCP9, GCP-WD, and all gene-coding augmin complex proteins were not identified. Single sequences were found as CKAP5 and TPX2. We collected all data about base known structural components of eukaryotic MTOC (according to [20,22,32]) in Table 1 and marked its presence in the diatom genomes and transcriptomes.

### 2.1. Gamma-Tubulin Complex Proteins—GCP

We have identified 127 predicted GCP amino acid sequences in 67 diatom species (GCP_fasta, Appendix A). Analysis of genome data of *Ulnaria acus*, *P. tricornutum*, *T. pseudonana*, *Pseudo-nitzschia multiseries*, and *Fragilariopsys cylindrus* revealed the presence of genes encoding GCP2 and GCP3. *Thalassiosira oceanica* has the only gene encoding GCP3 (Appendix A). Overall, GCP2 and GCP3 have been found in both centric and pennate diatom species (GCP2_fasta, Appendix A; GCP3_fasta, Appendix A). Analysis of transcriptomic data obtained from MMETSP allows one to find three types of GCP—GCP4, GCP5, and GCP6 (Appendix A)—and showed that, for 17 species, there are sequences of the one GCP isoform (Appendix A). Analysis of the domain structure revealed that many of the identified sequences are fragments, incomplete gene sequences. Data on the number of identified GCP sequences in large taxonomic groups of diatoms are presented in Table 2.

It has been shown previously that GCP2, GCP3, GCP4, GCP5, and GCP6 belong to a family of structurally related proteins [20,33]. While it is a well-known fact that all eukaryotes express the main components of MTOC, γ-tubulin, GCP2, and GCP3, the importance of all the additional components has remained unstudied [20,21,32]. The presence of even single GCP4, GCP5, and GCP6 sequences in representatives of Coscinodiscophyceae and Mediophyceae indicates that all GCPs can be identified in diatoms’ genomes.

In diatoms, GCP sequence identity within individual genera can be 40% in pennate diatoms (*Pseudo*-*nitzschia*), while, in centric species. sequences are more similar (63% identity in *Skeletonema* and *Thalassiosira*).

Diatom GCP2 length varied from 274 to 1698 a.a. which exceeds the GCP2 length of *H. sapiens* and *A. thaliana* (930 and 679 a.a., respectively), while *Phytophthora nicotianae* belonging to superphylum Heterokonta has similar GCP2 length (1210 a.a.; Appendix A). C-terminal domain structure is the most conserved; the domain part corresponding to 789–865 a.a. GCP2 of *H. sapiens* is enriched by lysine (K) and arginine (R), and the content of these polar positively charged amino acids reaches 26%. Sometimes they are represented by repeats from 2 to 4 K and R in different combinations. Earlier, such motives were described as potential signals for protein translocation to the nucleus [34,35]. Furthermore, it is known that the close relationship between MTOC and the nuclear envelope and its fragments during mitosis is characteristic of diatoms [24]. Probably, this relationship is carried out using the structure of some GCP domains.

The length of predicted amino acid GCP3 sequences varies from 221 to 1092 a.a. which is also longer than the sequences of the model organisms (838–938 a.a.; Appendix A). The identity between diatom GCP3s and well-studied GCP of other species is low and does not exceed 10%, while, among diatom sequence identity of the centric genus *Skeletonema*, it reaches 86%. The analysis of domain structure revealed that the most part of identified sequences are fragments of C- terminal part. For instance, there are repeats of K and R in C-terminal of centric species *Skeletonema* and *Thalassiosira* (3 and 6 a.a., respectively).

It is known that GCPs include two conserved motives, grip1 and grip2 located in N- and C-terminal parts, respectively [33,36]. Grip1 takes part in lateral contacts between GCPs, and grip2 provides connection with the γ-tubulin. Consensus amino acids for these motives are known for several well-studied species [36], but we found these motives and identified consensus amino acids for diatom GCPs (Figure 1). Diatom GCP3 grip1 motif has 4 substitutions in the consensus a.a. Grip2 motif is less conserved and has 17 substitutions in the consensus a.a.; these substitutions are characteristic for Phylum Bacillariophyta (Figure 2).

For the first time we identified genes encoding GCP4 in centric diatoms. The length of predicted amino acid GCP4 sequences varies from 272 to 733 a.a. Diatom GCP4 has low similarity (less than 10%) with GCP4 amino acid sequences of model organisms (GCP4_fasta, Appendix A). Moroever, diatom GCP4 sequences are more variable relative to each other than GCP2 and GCP3 (identity is no more than 15%). We could not find conserved blocks and patterns in the structure of predicted proteins. Analysis showed that the consensus a.a. of grip-motives also are not conserved (Figure 2). As we identified for diatoms, there is only one sequence of GCP5 (GCP5_fasta, Appendix A) and GCP6 (GCP6_fasta, Appendix A), which are small fragments of 359 and 530 a.a., respectively; this does not allow us to perform comparative analysis of amino acid sequences.

We study the post-translational modification sites of GCP2, GCP3, and GCP4 among *H. sapiens* and diatom sequences. For these proteins, acetylation, mono-methylation ubiquitination, and phosphorylation sites are characteristic. It was shown that sites of acetylation and mono-methylation are not conserved for diatom GCP2. Of 12 ubiquitination sites known for humans and other organisms, only two were represented in diatom sequences relevant to K449 and K455. However, according to the alignment, four conserved lysines were found—K377, K545, K677, and K778. The high conservation level of these amino acids in sequences of different diatom taxa allow one to assume that these lysines provide some functions and may be potential sites of post-translational modification including ubiquitination.

Protein phosphorylation is a key mechanism for activation and signal transduction. For GCPs, phosphorylation sites known for model organisms are not found in diatom sequences. Since most phosphorylation occurs at serine (S), threonine (T), and tyrosine (Y), we undertook a search for these conserved residues in diatom sequences. Three conserved residues Y were found in the diatom GCP2—Y484, Y547, and Y724. In addition, conserved S655 and T707 were found. Phosphorylation is possible at histidine (H) in bacteria, fungi, and plants [37,38]. We found five conserved histidines in diatom GCP2 sequences; all of them are located in the C-terminal domain—H572, H672, H690, H716, and H754—and could be potential sites of GCP2 phosphorilation (Appendix A).

For human GCP3, 42 sites for post-translational mono-methylation, ubiquitination and phosphorylation are known, but diatom GCPs do not have a single conserved one. Often arginine residues are located at the ubiquitination sites at lysine residues in diatom sequences. Putative ubiquitination sites may be conserved K563 and K761 at the diatom GCP3 C-terminal domain. Furthermore, ten conserved H and three conserved Y may be potential phosphorylation sites in the C-terminal domain (Appendix A). The conservation of amino acids for different diatom taxa including distantly related ones indicates the functional significance of these amino acids.

GCP4s were found in a small number of diatom species, which makes it impossible to reveal their patterns. We have not identified conserved amino acids that can be putative sites of post-translational modification. Mediophyceae diatoms have conserved histidine corresponding to human GCP4 H561 and S635 that we can suggest as potential diatom GCP4 post-translational modification (Appendix A). Earlier, other significant GCP4 amino acids were described in the N- and C-terminal domain [33]; of six conserved residues from the N-terminal domain, four were found in diatom predicted sequences—W190, D198, E202, and F203 (Appendix A). Six out of ten conserved residues from the C-terminal domain were found (Y456, P461, Y474, F478, H560, L564; Appendix A).

Thus, we assume that phosphorylation and ubiquitination are characteristics of diatom GCP post-translational modification; however, these sites do not correspond to human GCP. For every diatom GCP group, we identified several conserved Ks, Hs, and Ys, and these potential sites are located only in the C-terminal domain unlike other well-known model organisms. Perhaps this is due to features of complex formation between diatom γ-tubulin and GCPs and their activity regulation during cell cycle. We have not found any differences between centric and pennate potential post-translational modification sites, possibly because of their similar regulation.

Phylogenetic reconstruction of the GCP of model organisms and diatoms shows that GCP2, GCP3, and GCP4 form separate groups (Figure 3). At the same time, there is a division of GCP3 according to the centric and pennate species; the GCP2 region has a more polyphyletic structure; however, a clustering of centric and pennate sequences is noted. GCP2 and GCP3 outgroups (*H. sapiens*, *A. thaliana*, *S. cerevisiae*, *P. nicotianae*) are not grouped together with diatoms, emphasizing the specificity of each diatom. Thus, using the example of the GCP2 and GCP3 group distribution, we can assume that the evolution of the diatom GCPs is similar to the evolution of other elements of the cytoskeleton, such as EB proteins and tubulins, in these organisms [31,39].

### 2.2. Aurora A

Protein-kinases Aurora are a recently discovered family of kinases (with isoforms A, B, and C) that are involved in many mitotic events and are localized in the centrosome [40] and promote protein phosphorylation in mitosis [41]; they also participate in many mitotic events, such as regulation of the spindle assembly checkpoint, maintenance of centrosome and cytoskeletal functions. We identified 65 nucleotide sequences of serine/threonine protein kinase Aurora A for 55 diatom species.

Two genes encoding Aurora A have been identified for six diatom species. We assume that diatoms are characterized by only one isoform of this protein. The presence of two genes for some species may be due to compilation errors in the transcriptome analysis results, since all repeat sequences for six species were found in the MMETSP database. In the published genomes, the sequences of only one isoform of Aurora A have been found for each species. The identity of the predicted amino acid sequences in diatoms of different taxa can vary significantly (20–60%) and increases up to 95% within genera; for example, it can reach 99% for *Skeletonema*. Sequences have an average length of about 310–360 a.a. Aurora A consists of three domains, more variable N- and C-terminal regions and the catalytic kinase domain. The diatom Aurora A structure has similar features described for model organisms. Kinase domain has been identified for all diatom sequences and has a length of about 250–348 a.a. Homology within the kinase domain is high both between diatoms and model organisms (up to 68%). The N-terminal domain is more variable; its structure identity is observed only among several genera (Appendix A); for example, among species of the genus *Skeletonema*, it is almost completely identical. The C-terminal domain has a small length (6–10 a.a.) among most of the presented sequences; however, in some sequences its length can reach 130 and 365 a.a.

It is known that Aurora A catalytic activity is regulated by phosphorylation of the tyrosine (T288/T287) in its kinase domain and by association with other proteins [42]. The analysis of these and other modification sites was performed. It was shown that only T288 is conserved (Appendix A). For human Aurora A, 16 sites of post-translational modification are identified; we showed that there is characteristic S278 and T288. Other Aurora isoforms contain conserved residues T232 (AurB) and T195 (AurC) in the C-terminal part; their phosphorylation induces a conformational change associated with the acquisition of kinase activity [43,44]. Diatom Aurora A predicted amino acid sequences have conserved T at homological positions that can support their significance in diatom Aurora A activation (Appendix A).

Active unphosphorylated protein can bind with Mg^2+^ and asparagine 274 (D) [45]. Alignment analysis has showed that, in diatom sequences, a region homological to D is conserved. The phosphorylation of T287, S283, and S284 activated the kinase but inhibited it under S342 modification. However homologous these amino acids are in diatom, the predicted sequences are not conserved (Appendix A). Human Aurora A can be regulated by Ca^2+^/calmodulin binding that induces its autophosphorylation at S51, S53/54, S66/67, and S98. These residues are not conserved either; that was predictable since they are all located in the variable N-terminal domain.

In mitosis, Aurora A is activated and autophosphorylated at residue T288 [46,47]. A consensus motif for one of the Aurora A phosphorylation substrates is described as R/K/N-R-X-S/T-B, where B denotes any hydrophobic residue, with the exception of P, at which autophosphorylation of Aurora A can occur [48]. In diatom Aurora A, this motif is identified only in region T288 and has the general formula R-R-N-T-L. This fact allows us to state that the structurally catalytic domain of Aurora A in diatoms is similar to that of other organisms, and its activation occurs due to the autophosphorylation of T288 as part of the R-R-N-T-L sequence. Thus, based on the analysis, we assumed that only few conserved amino acids are significant for the regulation of Aurora A in diatoms and located in the C-terminal part of the kinase domain.

The interaction between centrins and Aurora A, which provides phosphorylation of centrins and is localized together with them in the centrosome, has been well studied [40]. According to the recent studies [40,48,49], Aurora A phosphorylates’ centrin at serine residues (S) 170 and 122 in the presence of ATP. Both of these sites (120KISF123 and 168KTSL171) correspond to the consensus substrate sequence for Aurora A (R/K-x-T/S-I/L/V/F or R/K/N-R-X-S/T-B) and are also found in diatom centrins (Appendix A).

### 2.3. Centrins

In some unicellular eukaryotes (e.g., *S*. *cerevisiae* and *Chlamydomonas reinhardtii*), one gene-encoding centrin was found [50,51], while in others, more than two dozen genes were identified (e.g., in *Paramecium caudatum*, there were found 22 genes, and in *Trichomonas vaginalis*, 24 genes [52]). These proteins are characterized by the presence of EF-hands motifs; each consists of two 12 amino acid EF-hand motifs that can potentially bind Ca^2+^.

We identified 42 centrin sequences in 22 diatom species. It is noteworthy that most of the sequences (28) belong to the Mediophyceae. Nine species have two or more sequences that can be characterized as centrins. The largest number (eight) of amino acid sequences of centrins was found for the species *Pseudo*-*nitzschia fraudulenta*; however, some of them have a length of less than 110 a.a. and represent only a fragment of the amino acid sequence. Sequence identity with the reference group is more than 40% and, within members of the Mediophyceae, is more than 52%, which is consistent with the literature, in which centrins are described as proteins with high homology [53]. The length of the predicted amino acid sequences ranges from 104 to 232 a.a.; thus, some of the sequences are not complete, and we found that they contain only one EF-hand domain pair or their incomplete construction (Appendix A).

Previously, comparison of *Naegleria gruberi* centrins [54] with sequences of other organisms, including the green alga *Chlamydomonas reinhardtii*, revealed conserved amino acids common to organisms from different taxonomic groups. Since the analysis of diatom genomes showed an abundance of genes related to green algae [55,56], a search for conserved amino acids was carried out. The Coscinodiscophyceae and Bacillariophyceae are represented by single sequences; however, the analysis of the sequences of the Mediophyceae made it possible to detect most of the conserved a.a. (Appendix A).

Interestingly, diatom centrins in the N-terminal part have a positively charged region enriched in lysine (K) and arginine (R) residues. It is suggested that this region may be significant for Ca^2+^-induced polymerization [57]. However, the location of K and R in this area is only conserved within the genus, for instance, among *Thalassiosira* and *Skeletonema* (centrins.fas). Post-translational modifications regulate the activity and subcellular localization of centrins, and, as expected, this may allow organisms with relatively fewer centrins to realize their diversity of functions, similar to organisms with a large number of centrins [58]. Analysis of the sites of post-translational modification showed that putative ubiquitination (K78, R111, K120, K127, K167) and phosphorylation (T26) sites retain their conservation.

All identified diatom centrin sequences contain two EF-hand domain pairs or their fragments (Appendix A) separated by linker regions. The DxDxDG sequence within each EF-hand motif ensures high-affiliation binding of Ca^2+^ ions [59]. Analysis of the structure of diatom sequences showed that it is retained only in the fourth EF-hand motif, while in the other three motifs, among all identified sequences, it can acquire the structure shown in Table 3. Interestingly, glycine (G) at the end of this sequence, which provides interaction with Ca^2+^, remains conserved in all four motifs. In the third EF-hand motif, the sequence is conserved in diatoms of different taxa and is represented by the DDDETG sequence, while, in the other three EF-hand motifs, it is more variable (Table 3; Appendix A). Earlier, when analyzing the structure of centrins, it was noted that the 12th a.a. of the EF-hand motif is glutamate (E). The replacement of glutamate by aspartic acid (D) is not uncommon; however, as the authors note, this may cause a decrease in the selectivity of Ca^2+^-binding in favor of binding Mg^2+^ ions [59,60]. It was shown that the E at this position is conserved in first, second, and fourth EF-hand motifs, while asparagine (N) occupies this position in the third EF hand motif. It is possible that the third motif is not involved in the binding to Ca^2+^ ions.

We have identified only 42 centrin sequences, most of which belong to the class Mediophyceae, so it is difficult to judge the evolution of centrins in different classes of diatoms based on phylogenetic reconstruction due to a lack of data (Figure 4).

## 3. Materials and Methods

### 3.1. Search of Sequences

The search for the sequences of GCP, protein kinase Aurora A, centrins, proteins of Augmin complexes, and components providing independent nucleation without MTOC (CKAP5 and TPX2) was performed in the set of amino acid sequences predicted on the basis of published genomic data of *Ulnaria acus* (Kützing) by Aboal, *Synedra acus* subsp. *radians* (Kützing) by Scabitchevsky [61], *Phaedactylum tricornutum* by Bohlin [62], *Thalassiosira pseudonana* by Hasle and Heimdal [56], *Thalassisosira oceanica* by Hasle [63], *Pseudo-nitzschia multiseries* (Hasle) by Hasle, *Fragilariopsis cylindrus* (Grunow) by Krieger [64], and transcriptome data presented in MMETSP [65] using blast+ [66]. As a query, the sequences of *Homo sapiens*, *Arabidopsis thaliana*, *Saccharomyces cerevisiae* were used (Table 4). E-value threshold was 1 × 10^−10^.

All identified sequences were tested by blasting the NCBI database nr [67] to remove bacterial sequences and other contaminants and using hmmer 3.3.2 to confirm that their domain structure matches the expected structure.

### 3.2. Alignment and Comparative Amino Acid Sequence Analysis

Alignments for structural and phylogenetic analyzes were made using Clustal Omega 1.2.4 [68] and corrected manually using aliview [69]. The search for functionally significant domains was performed using the InterPro [70]. All amino acid positions are numbered according to the positions in the sequences of the studied *H. sapiens* proteins with which they coincide. PhosphoSitePlus v6.6.0.4 data were used to search for and analyze post-translational modification sites in the predicted amino acid sequences. The search for grip-motifs in the structure of GCP and DxDxDG motifs in centrins was carried out using previously published information [33,36,42,54,58,71].

### 3.3. Phylogenetic Analysis

For GCP phylogenetic analysis protein sequences, protein sequences were aligned using muscle (v3.8.31; [72]) with default parameters. Alignment was filtered by trimAI (v. 1.2; [73]) to remove columns carrying gaps in more than 30% of protein sequences (parameter -gt 0.7). To find the best threshold as a compromise between length of the resulting alignment and its quality, we performed tests with the parameter -gt varying from 0.3 to 0.9, and parameter value 0.7 was chosen as the most suitable. After this procedure, there were 144 protein sequences and 592 positions in alignment left. Alignment quality control was performed in UGENE (v. 46.0; [74]). Aligned and filtered sequences were converted to phylip format for subsequent phylogenetic analysis. Phylogenetic reconstruction was performed using RAxML (v.8.0.26; [75]) with a model of rate heterogeneity GAMMA, amino acid substitution model WAG, and 100 bootstrap iterations giving a consensus tree. Visualization was performed in iTOL (v. 6.7.4; [76]). Computational resources of the “Makarich” HPC cluster were provided by the Faculty of Bioengineering and Bioinformatics, Lomonosov Moscow State University, Moscow, Russia.

For phylogenetic analysis of centrins, the Maximum Likelihood method was performed with IQ-Tree v1.6.12 [77]. The replacement model was selected using the built-in ModelFinder algorithm [78]. Individual branch support was assessed using 1000 bootstrap replicas for each tree obtained by the UFBoot2 algorithm [79]. Trees were rendered in FigTree v1.4.4.

## 4. Conclusions

In this study, sequences of different diatom taxa—the MTOC components GCP2, GCP3, GCP4, GCP5, and GCP6—were identified. All systematic groups are characterized by the presence of genes encoding GCP2 and GCP3 in their genomes. Both classes of proteins in the predicted sequences have conserved amino acids characteristic only for diatoms, which indicates their importance for this taxon. Since the diatom consensus amino acids of grip-motifs are similar to those in other organisms, it can be assumed that the mechanisms of GCPs binding to each other and to γ-tubulin also have much in common. Previously, GCP4–6 were not found in diatoms; however, we were able to identify GCP4 sequences in centric diatoms. The existence of diatoms GCP5 and GCP6 has not yet been confirmed, and, despite the fact that in this study we found only single sequences for them, it might be speculated that these genes are present in diatom genomes. This observation will be investigated in further studies. 

It was possible to establish the high conservation level of several a.a., which may be the sites of post-translational modification, ubiquitination, and phosphorylation. In contrast to model organisms, where the sites are located along the entire length of the sequences, in diatoms, they are concentrated in the C-terminal domain. It is possible that this is due to the peculiarities of the formation of the complex between γ-tubulin and these proteins, the regulation of their activity, and the limitation of the protein activity at certain stages of the cell cycle. There are no differences in potential modification sites between centric and pennate species, which allows us to conclude that GCP in each class of diatoms has a similar regulation.

In the diatoms’ Aurora A-predicted sequences, the general patterns of the protein kinase structure described for model organisms are observed. T288 is conserved in all identified sequences. It is noteworthy that it is a part of the motif that serves as a substrate for the Aurora A activity self-regulation through autophosphorylation. This motif is conserved in evolutionarily distant species and, therefore, is necessary for protein function. 

The analysis of centrins showed that these proteins are conserved for diatoms and other organisms. The presence of R/K-x-T/S-I/L/V/F or R/K/N-R-X-S/T-B motifs which are substrates for phosphorylation by Aurora A indicates the existence of regulatory pathway centrins with Aurora A. The conservation of the third EF-hand motif of centrins in different diatom classes may indicate in favor of a common ancestral form of the protein, and its preservation in both older and younger species.

The special sequence structure studied in this work in representatives of the genera *Thalassiosira* and *Skeletonema* (radial centric species from the family Thalassiosiraceae and Skeletonemataceae) attracts special attention. Previously, the same fact was noted for EB-proteins for these genera [39]. Both genera systematically belong to the order Thalassiosirales; however, there are no data on other representatives of the order, and the questions arise whether they will have the same features and what was the reason for their occurrence, clearly distinguishing them from others among the centric ones. Thus, for the first time in the body of work, the structures of the main participants of the MTOC were analyzed in detail, and the feature characteristics of diatoms were revealed.

## Figures and Tables

**Figure 1 ijms-24-12781-f001:**
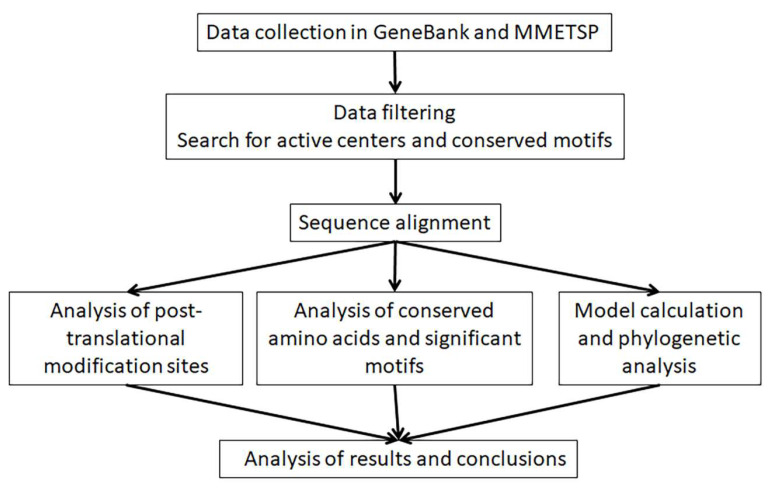
Scheme reflecting the order of the study.

**Figure 2 ijms-24-12781-f002:**
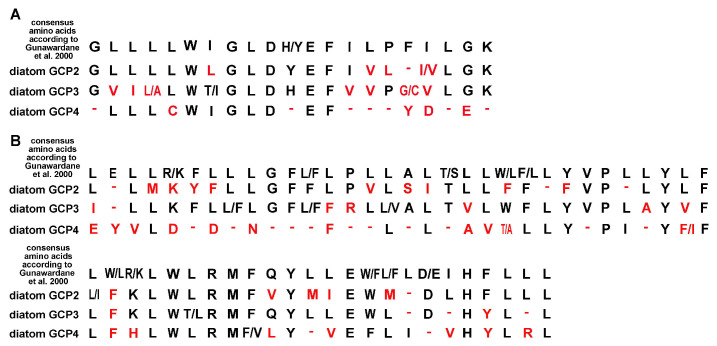
Consensus a.a. of GCP grip-motives for diatoms. For comparison, consensus a.a. according to [36] are shown. (**A**) grip1 motif; (**B**) grip2 motif. Red indicates amino acids of diatoms that differ from the consensus amino acids of other model organisms.

**Figure 3 ijms-24-12781-f003:**
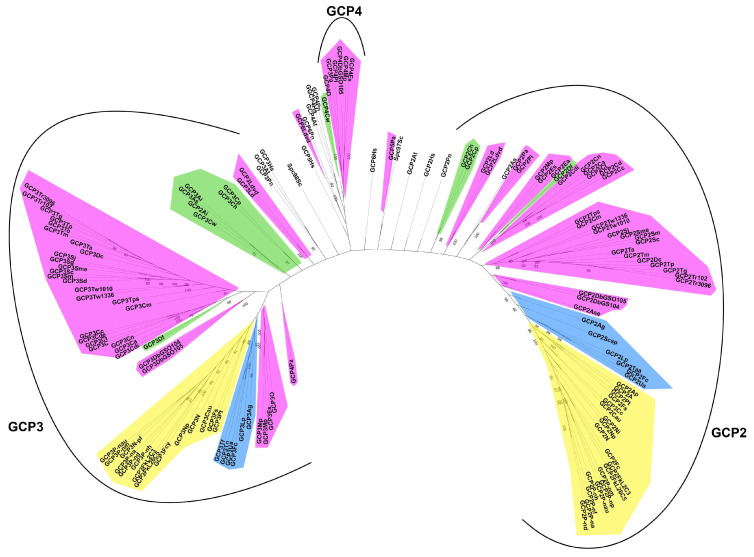
A phylogenetic network of GCP predicted amino acid sequences. Major taxonomic groups (classes and subclasses) are highlighted in colors. Annotations: purple—Mediophyceae; green—Coscinodiscophyceae; blue—Bacillariophyceae, Urneidophycidae, and Fragilariophycidae; yellow—Bacillariophyceae, Bacillariophycidae.

**Figure 4 ijms-24-12781-f004:**
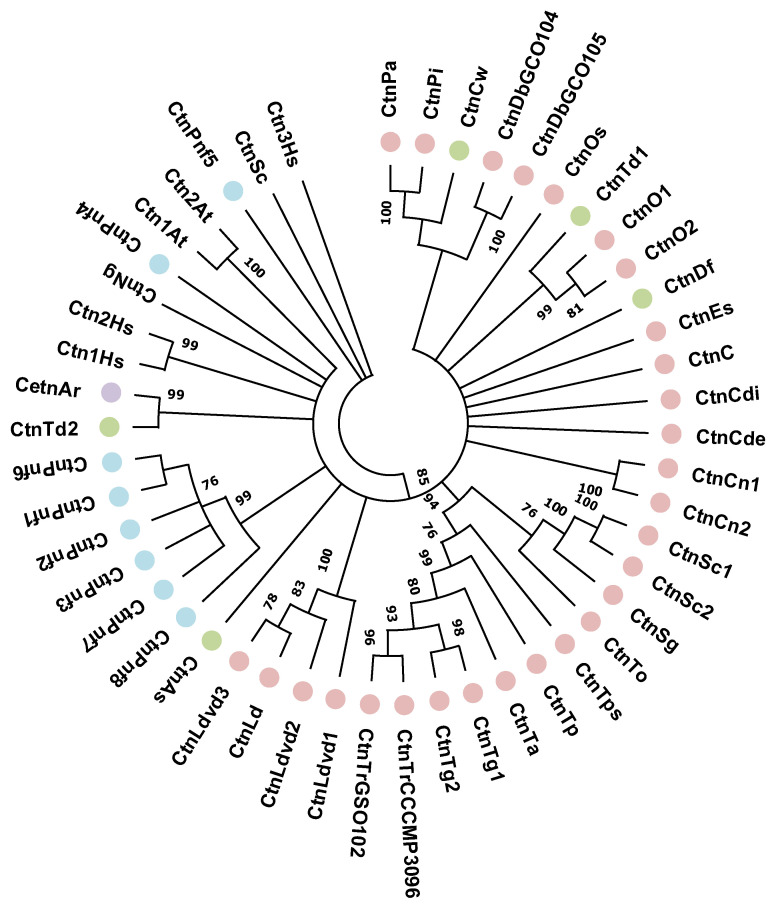
Phylogenetic reconstruction of diatom centrins. Coscinodiscophyceae are marked with green circles, Mediophyceae with pink circles, Bacillariophyceae and Fragilariophycidae with blue circles, and Bacillariophyceae and Bacillariophycidae with purple circles.

**Table 1 ijms-24-12781-t001:** Structural components of MTOC of various organisms, according to [20,21,22] and this study.

MTOC Components	*Homo* *sapiens*	*Arabidopsis thaliana*	*Saccharomyces* *cerevisiae*	*Phytophthora* *nicotianae*	Diatoms
GCP2	+	+	+	+	+
GCP3	+	+	+	+	+
GCP4 (*TUBGCP4*)	+	+	–	+	+
GCP5 (*TUBGCP5*)	+	+	–	–	+/−
GCP6 (*TUBGCP6*)	+	+	–	–	+/−
GCP8A (*MZT2A*)	+	+	–	–	−
GCP8B (*MZT2B*)	+	+	–	–	−
GCP9 (*MZT1*)	+	+	–	+	−
GCP-WD (NEDD1)	+	+	–	+	−
Protein kinase Aurora A	+	+		+	+
Centrins	+	+	+	–	+
Nucleolins	+	+	+	–	+
Augmin complex proteins	+	+	–	–	−
CKAP5	+	+	–	–	+/−
TPX2	+	+	–	–	+/−

“+” homologous sequences found in the genome; “–” no data; “+/−” found a small number of sequence fragments in the genomes.

**Table 2 ijms-24-12781-t002:** Distribution of the identified sequence numbers among large taxa in diatoms.

	Species	GCP2	GCP3	GCP4	GCP5	GCP6
Coscinodiscophyceae	6	5	6	1	1	1
Mediophyceae	35	32	31	6	-	-
Bacillariophyceae, Urneidophycidae	1	1	1	-	-	-
Bacillariophyceae, Fragilariophycidae	7	4	5	-	-	-
Bacillariophyceae, Bacillariophycidae	18	17	16	-	-	-
Bacillariophyta		59	59	7	1	1

“-“ no homologous sequences found.

**Table 3 ijms-24-12781-t003:** Structure of the DxDxDG sequence in EF-hand motifs from different diatom classes and subclasses.

	EF-Hand Domain Pair 1		EF-Hand Domain Pair 2	
	EF-Hand 1	EF-Hand 2	EF-Hand 3	EF-Hand 4
Bacillariophyta	DxDxDG	DxDxDG	DxDxDG	DxDxDG
Coscinodiscophyceae	DXDGXG	DKDGSG	DDDETG	DXDGDG
Clas Mediophyceae	DTDGSG	DKDGXG	DDDETG	DXDGDG
Bacillariophyceae Urneidophycidae	–	–	–	–
Bacillariophyceae Fragilariophycidae	DTDGSG	DDDGXG	DDDETG	DXDGDG
Bacillariophyceae Bacillariophycidae	DTDGSG	DDDGSG	DDDETG	DXDGDG

“–“ no data.

**Table 4 ijms-24-12781-t004:** Sequences used as query and their accession numbers in GeneBank.

	*Homo sapiens*	*Arabidopsis thaliana*	*Saccharomyces cerevisiae*	*Phytophthora* *nicotianae*	*Ectocarpus siliculosus*	*Naegleria* *gruberi*
GCP 2	NP_001243546	AED92422	QHB09163	KUF64340		
GCP 3	NP_006313	Q9FG37	AJT23326	KUF76510		
GCP 4	NP_001273343	OAP01290		KUF76976		
GCP 5	AAK77662					
GCP 6	NP_065194			KUG00998		
Protein kinase Aurora A	A—NP_001310232.1 B—AAH00442 C—KAI4045108.1	1—OAP01184.1 2—OAP10838.1 3—OAP01290	PJP07804.1	A—KUF82775.1 A-A—KUF81362.1 A-B—KUF80014.1	CBN77021.1	CtnNg—XP_002671077.1
centrin	Ctn1Hs—NP_004057.1 Ctn2Hs—CAA51467.1 Ctn3Hs—KAI4021980.1	Ctn1Hs—sp|082659.1 Ctn2Hs—NP_190605.1	CtnSc—PJP11597.1			

## Data Availability

No new data were created, all the necessary links to sites and sequence numbers are contained in the materials and methods of the paper and in Appendix A.

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
