# Peer review of "Analysis of Predicted Amino Acid Sequences of Diatom Microtubule Center Components"

_ijms, 2023, doi:10.3390/ijms241612781_

Round 1
Reviewer 1 Report
Several MTOC in diatoms components, including GCP (gamma complex proteins), Aurora A, and centrins, have been found in the current research. These proteins' projected amino acid sequences underwent analysis, and the results showed structural characteristics common to diatoms. The authors looked at the conserved amino acids and the factors required for proteins to function. According to their systematic location, all main groupings of diatoms are dispersed along phylogenetic trees, according to phylogenetic analysis of GCP. The work is generally well written, the techniques are well explained, and the results are thoroughly discussed. In particular the data presented in this paper will be a valuable addition to the literature. I recommend the paper the paper for publication after addressing my minor concern.
Since the study is fully computational, I request the author to make a statement of study limitation in the abstract/conclusion section of the manuscript.
Add a flowchart to schematically represent the overall methodology of the manuscript to enhance the readership.
Author Response
We are grateful to the reviewer for the work done.
According to the suggestion of the reviewer, a figure with a study scheme has been added (designated as Figure 1), a reference to the figure in the text has been added, the numbering of the figures has also been corrected.
Statement was added to abstract “This work is a theoretical study; however, it allows drawing some conclusions about the functioning of the studied components and possible ways of their regulation.”

Reviewer 2 Report
I find the paper well structured and presented. The introduction is well written and give a good idea of the state of the art of the issue analysed, the methods are precise and complete, describing each section in an appropriate way and the results describe the analyses were performed in depth giving a complete picture of the study. The references resulted appropriate and recent. The tables and figure are clear and well representing data without any redundance. The scientific interest is in the average and it has been developed in a rationale and analytic way.
Author Response
We are grateful to the reviewer for the work done.